# ATPase Thorase Deficiency Causes α-Synucleinopathy and Parkinson’s Disease-like Behavior

**DOI:** 10.3390/cells11192990

**Published:** 2022-09-26

**Authors:** Fei Gao, Han Zhang, Jia Yang, Menghua Cai, Qi Yang, Huaishan Wang, Yi Xu, Hui Chen, Yu Hu, Wei He, Jianmin Zhang

**Affiliations:** 1Department of Immunology, Research Center on Pediatric Development and Diseases, Key Laboratory of T Cell and Immunotherapy, Institute of Basic Medical Sciences, Chinese Academy of Medical Sciences and School of Basic Medicine, Peking Union Medical College, State Key Laboratory of Medical Molecular Biology, Beijing 100005, China; 2Changzhou Xitaihu Institute for Frontier Technology of Cell Therapy, Changzhou 213000, China; 3Haihe Laboratory of Cell Ecosystem, Chinese Academy of Medical Sciences and Peking Union Medical College, 288 Nanjing Road, Tianjin 300020, China

**Keywords:** Thorase, α-synuclein, Parkinson’s disease

## Abstract

Parkinson’s disease (PD) is one of the most common neurodegenerative diseases and is pathologically characterized by α-synucleinopathy, which is harmful to dopaminergic neurons. However, the underlying mechanisms and pathogenesis of PD remain unclear. The AAA + ATPase Thorase was identified as being essential for neuroprotection and synaptic plasticity by regulating the AMPA receptor trafficking. Here, we found that conditional knockout of Thorase resulted in motor behaviors indicative of neurodegeneration. Genetic deletion of Thorase exacerbated phenotypes of α-synucleinopathy in a familial PD-like A53T mouse model, whereas overexpression of Thorase prevented α-syn accumulation in vivo. Biochemical and cell cultures studies presented here suggest that Thorase interacts with α-syn and regulates the degradation of ubiquitinated α-syn. Thorase deficiency promotes α-syn aggregation in primary cultured neurons. The discoveries in this study provide us with a further understanding of the pathogenesis of α-synucleinopathies including PD.

## 1. Introduction

Parkinson’s disease (PD) is the second most common neurodegenerative disease and is characterized by motor symptoms (bradykinesia, rigidity, postural instability, and rest tremor) and some non-motor symptoms (such as cognitive impairment of sensory alterations) [1]. However, no efficacy approaches are currently available to treat or prevent the patient’s physical and mental health declines from a certain period after the onset of the disease. PD is also pathologically characterized by α-synucleinopathy which is harmful to dopaminergic neurons. α-synuclein(α-syn) aggregation causes the formation of Lewy bodies (LB) which triggers a cascade of events leading to eventual neuronal cell death [2].

The etiology of PD is involved in genetic factors for familial PD and environmental factors for sporadic PD. The first evidence of the α-syn aggregation/PD relationship was first discovered in a large Italian and three Greek families with a point mutation in the SNCA gene encoding α-syn [3]. In addition, several other genes have also been identified to be involved in the development of autosomal and recessive inheritance of PD, such as LRRK2, PINK1, VPS35, GBA, PARK2, PARK7, DJ1, UCHL1 [4].

Previous studies have demonstrated that Thorase is neuroprotective and controls synaptic plasticity by regulating the AMPA receptor trafficking [5,6]. Physiological studies showed that genetic ablation of Thorase in dopamine neurons resulted in impaired induction of synaptic plasticity, and animals lacking Thorase in DAT+ neurons expressed greater associative learning in a fear conditioning paradigm [7]. Recently, Thorase mutations were linked to schizophrenia [8], congenital immobility, stiffness [9], and lethal encephalopathy [10], as well as a devastating neurological disorder characterized by hypertonia, seizures, and death in a consanguineous family [11].

In this study, we identified Thorase as a key mediator regulating α-syn metabolism. Thorase conditional knockout (cKO) mice developed spontaneous pathologies and motor dysfunction behavior mimicking PD and α-synucleinopathy.

## 2. Materials and Methods

### 2.1. Mouse Handing

The Thorase^+/−^ mice, Thorase^flox/+^ mice, and conditional Thorase transgenic (cTg) mice were gifted by Drs. Ted Dawson and Valina Dawson from the Institute of Cell Engineering, Johns Hopkins University School of Medicine [5]. Conditional Thorase knockout mice (cKO) were generated by crossing Thorase^flox/+^ mice with CaMKIIa-iCre [12]. B6.Cg-2310039L15Rik^Tg(Prnp-SNCA*A53T)23Mkle^/J transgenic mice (referred to as hA53T) [13] were obtained from the Jackson Laboratory (Bar Harbor, ME, USA). Thorase cKO-hA53Tα-syn mice and Thorase cTg-hA53Tα-syn mice were also generated by a two-round crossing of Thorase cKO mice and Thorase cTg mice with hA53T transgenic mice, respectively. All mouse breeding and handling procedures were approved by the Institutional Animal Care and Use Committee (IACUC) at the Institute of Basic Medical Sciences (IBMS) of the Chinese Academy of Medical Sciences (CAMS) and the School of Basic Medicine (SBM) of Peking Union Medical College (PUMC) (Approval No. 81271415).

### 2.2. Mouse Genotyping

The genotypes of mice were determined by PCR of DNA obtained from tail screening. The primers for mouse genotyping were as follows:

Primers for genotyping Thorase^−/−^ and Thorase^flox/+^: JZ129 5*′*-GCA TGT GCA ACT GAA CGC AGG G-3*′*, JZ156 5′-CCA CCC ACT TTG TTT TGA AGT TAA-3*′*, and JZ178 5′-GGA GCT TTC AGT CAG TAC CAT TTC ACT TTC-3*′*.

Primers for genotyping CaMKIIa-iCre: JZ179 5′-GAC AGG CAG GCC TTC TCT GAA-3*′*, JZ180 5′-CTT CTC CAC ACC AGC TGT GGA-3*′*;

Primers for genotyping Thorase cTg: JZ258 5′-CGG GTC GAG TAG GCG TGT AC-3*′*, JZ259 5′-G ATCCA TGA TCC CCG GGT ACC GAG-3*′*, JZ262 5′-TGA AAG TGG GTC CGC GTA C-3*′*, and JZ263 5′-TAC TCG TCA ATT CCA AGG GC-3*′*.

Primers for genotyping hA53T: OMIR1772 5′-TGT AGG CTC CAA AAC CAA GG, OMIR3560 5′-TGT CAG GAT CCA CAG GCA TA 3*′*, OMIR7338 5′-CTA GGC CAC AGA ATT GAA AGA TCT-3*′*, and OMIR7339 5′-GTA GGT GGA AAT TCTA GCA TCAT CC-3*′*.

### 2.3. Primary Hippocampal Neuron Culture

Primary hippocampal neuron cultures were prepared from early postnatal (P0–P1) mouse brains and maintained as described previously [14]. Neurons were cultured in Neurobasal TM medium (Thermo Fisher Scientific, Waltham, MA, USA) with supplemental B27 (Thermo Fisher Scientific) and 0.5 mM GlutaMAX™-I (Thermo Fisher Scientific) during the first 5 days. Subsequently, neurons were cultured in Neurobasal TM medium without GlutaMAX™-I, and the culture medium was changed every two days.

### 2.4. Preparation of Human a-syn-A53T PFFs

The pRK172 plasmid harboring the gene for full-length human α-synuclein was purchased from Addgene. The PD-associated A53T mutation was achieved by using a QuikChange Lightning Site-Directed Mutagenesis kit (Agilent Technologies, Santa Clara, CA, USA) according to the instruction manual. The oligonucleotides for A53T were forward = 5′-GAGTGGTGCATGGTGTGACAACAGTGGCTGAG-3′’ and reverse = 5′-CTCAGCCATG TTGTCACACCATGCACCACTC-3′. The A53T mutation plasmid was validated by Sanger sequencing.

α-Syn PFFs were generated from recombinant mutant A53T α-syn, as described previously [15]. The hA53T-PFFs were diluted in PBS at 0.1 mg/mL, sonicated with 30 pulses at 10% power and added to neuronal media with 5 μg/mL of hA53T PFFs in a 24-well tray [5,6,16]. On DIV 7, neurons were co-cultured with PFFs for 48 h before the removal of culture supernatant and incubation with 500 μL of 4% paraformaldehyde at room temperature (RT) for 15 min. After washing three times with PBS for 5 min, the coverslips were permeabilized with 0.1% Triton X-100 at RT for 20 min and then blocked with 5% BSA in PBS for 2 h.

To examine the detergent insoluble α-syn aggregates, coverslips were incubated with primary antibodies at 4 °C overnight before 5 times wash with 0.1% TBST and the incubation with Alexa-Fluor 555-labeled anti-Mouse or Alexa-Fluor 647-labeled anti-Rat secondary antibody at RT for 1 h. Then the coverslips were placed upside down on the new glass slides with ProLong Gold Antifade Mountant (Thermo Fisher Scientific) with DAPI. Images were acquired by using a Zeiss LSM 780 confocal microscope and analyzed by Zeiss ZEN Service Blue software.

### 2.5. Immunoblot Analysis

Brain tissues were homogenized in RIPA buffer (10 mM Tris-HCl, pH 7.4, 150 mM NaCl, 5 mM EDTA, 0.5% sodium deoxycholate, 1% Triton X-100, 1.5% SDS) containing protease and phosphatase inhibitors (Thermo Fisher Scientific) on ice. Lysates were sonicated and centrifuged for 20 min at 4 °C and 15,000× *g*, and the supernatants were collected. Triton X-100 soluble and insoluble fractions were acquired by homogenization of brain tissues in Triton X-100 lysis buffer (10 mM Tris-HCl, pH 7.4, 150 mM NaCl, 5 mM EDTA, 1% Triton X-100) containing protease and phosphatase inhibitors on ice. The homogenates were sonicated and centrifuged for 20 min at 4 °C, 100,000× *g*, and the resulting supernatants were collected as the Triton X-soluble fractions. Pellets were washed and re-lysed in SDS lysis buffer (10 mM Tris-HCl, pH 7.4, 150 mM NaCl, 5 mM EDTA, 2% SDS). The protein concentration was measured using a BCA kit (Thermo Fisher Scientific). Lysates (15–20 μg) were loaded on 8–12% SDS-PAGE gels for electrophoresis and then transferred to nitrocellulose (NC) membranes. The membranes were blocked in 5% milk or 5% BSA solutions for 1 h. The primary antibodies were incubated at 4 °C overnight. After washing 5 times with 0.1% PBST, the NC membranes were then incubated with HRP-labeled secondary antibodies (ZSGB, Beijing, China) at RT for 1 h, and washed as above. Chemiluminescent substrate was added (SuperSignal West Pico, Thermo Fisher Scientific,), and images were developed using ChemiScope, CLiNx Science Instruments (Shanghai, China). Gray values of the designated bands on the images were measured using ImageJ software.

### 2.6. Co-Immunoprecipitation Assay

HEK293 cells were transiently cotransfected with the plasmids cFUGW-Thorase-myc, cFUGW-α-syn-GFP, or cFUGW-GFP using jetPRIME Transfection Reagent (Polyplus Transfection, Illkirch, France) according to the instruction manual. The cells were washed twice with PBS 48 h after transfection and harvested with lysis buffer containing 25 mM Tris base, 150 mM NaCl, 1 mM EDTA, 0.5% NP-40, 1% Triton X-100, 0.5% deoxycholate and protease and phosphatase inhibitors (Thermo Fisher Scientific) on ice, and centrifuged at 20,000× *g* for 20 min. The supernatants were precleared with 10 μL of uncoupled agarose beads for 2 h and the supernatants were determined by using a BCA assay kit (Thermo Fisher Scientific). Precleared samples were separated into 500 μg of protein and incubated with GFP-Trap beads (Chromo Tek, Munich, Germany). The IP complexes were washed 5 times with wash buffer containing 25 mM Tris base, 300 mM NaCl, 1 mM EDTA, and 1% NP-40. Western blotting was performed to examine the interaction between Thorase with α-syn using an anti-GFP antibody.

In the in vivo assay, approximately 500 μg of protein lysate from adult mouse brain in lysis buffer was incubated with 1 μg anti-Thorase antibody or mouse iso-IgG as control at 4 °C overnight. An amount of 40 μL of Pierce Protein A/G Magnetic Agarose (Thermo Fisher Scientific) was added and incubated at room temperature for 2 h to recover the immunocomplexes with the aid of a magnetic stand. The immunocomplexes were then washed 4 times with wash buffer. Western blotting was then performed to examine the interaction of Thorase with α-syn by anti-Thorase antibody and anti-α-syn antibody [5].

### 2.7. Ubiquitination Assay

Ubiquitination assays were performed as previously described [16]. Briefly, Ub-HA and α-syn-GFP expression constructs were transiently cotransfected with or without cFUGW-Thorase-myc or pcDNA4/myc-His into HEK293 cells. Twenty-four hours after transfection, cells were incubated for 4 h with 5 mM MG132, 25 μM bafilomycin A1 (Selleckchem, Houston, TX, USA), or vehicle (DMSO, final concentration 0.1%). To inhibit protein synthesis, cells were treated with 100 μg/mL cycloheximide (MedChem Express, Monmouth Junction, NJ, USA). The cells were then harvested and lysed for immunoprecipitation using GFP-Trap beads. Western blots were performed to examine the immunocomplexes; an anti-HA antibody was used to detect ubiquitin conjugates.

### 2.8. Immunohistochemistry

Brain sample was fixed in 4% paraformaldehyde and incubated in 15% and 30% sucrose solutions and embedded in O.C.T. Compound (Tissue-Tek, Sakura, Japan). Coronal samples were cryosectioned (Leica CM1950, Bensheim, Germany) at 30 μm. Subsequently, sections were blocked with 5% goat serum and incubated with primary antibodies at 4 °C overnight and with biotinylated secondary antibodies (Vector Laboratories, Newark, CA, USA) at room temperature for 45 min. HRP was detected using a 3,3′diaminobenzidine (DAB) kit (ZSGB, Beijing, China). High-power and whole view sections were acquired using a Nikon microscope (Nikon, Tokyo, Japan). Images were analyzed with Image Pro-Plus 2.0 (Rockville, MD, USA).

Dopaminergic (DA) neurons from the SNpc region were counted using an optical fractionator. Briefly, brains were frozen, and serial coronal sections (40 μm) were mounted on glass slides. We performed continuously sliced brain tissue and collected it sequentially at 5 intervals. Then, we choose brain tissue sections in similar plane for experiments. Sections were permeabilized in 0.3% Triton X-100 and blocked in 5% goat serum at room temperature for 60 min. Sections were then incubated in rabbit polyclonal anti-TH antibody overnight at 4 °C. Sections were incubated for 45 min in HRP anti-rabbit IgG antibody and visualized with a DAB Kit (ZSGB). Unbiased stereological cell counts were collected using the optical fractionator method to count TH-positive neurons in every fifth SNpc section in a total of 6–10 sections at 20× magnification using Stereo Investigator version 6.0 software (MicroBright-Field, MBF Bioscience, Williston, VT, USA). The fiber density in the striatum was quantified by OD. ImageJ (NIH, Bethesda, MD, USA) was used to analyze the OD as previously described [16].

### 2.9. Immunofluorescence Staining

Brain sample was fixed in 4% paraformaldehyde and dehydrated in 15% and 30% sucrose solutions and embedded in O.C.T. Compound (Tissue-Tek, Sakura, Japan). The frozen samples were cut into slices using Leica CM1950 cryostats (Bensheim, Germany) about 30 μm thick. Subsequently, sections were blocked with 5% goat serum and incubated with diluted primary antibodies at 4 °C overnight. After washing 5 times with TBST buffer, slices were incubated with appropriate Alexa Fluor Secondary antibodies (Thermo Fisher, 1:2000) at room temperature for 45 min and covered with Prolong Gold Antifade Mountant (Thermo Fisher) with DAPI (Thermo Fisher). Images were taken on a Zeiss LSM780 confocal microscope in the same parameters between different groups. For post-processing, ZEN lite 2.3 SP1 software (White Plains, NY, USA) was used to adjust the images and add a scale. For average fluorescence intensity and fluorescence area within individual cells measuring, ImageJ software (NIH, Bethesda, MD, USA) was used for statistics.

### 2.10. Behavioral Measurements

Mice were acclimatized to the experimental arena daily for 5 days. All behavioral experiments were conducted in the light phase. The experimenter was blinded to the genotype of the mice. The apparatus and the testing surface was cleaned with 75% ethanol and allowed to dry before testing each animal to minimize odor cues.

#### 2.10.1. Open Field Activity

The open field test was performed to evaluate the general activity, locomotion, and anxiety of mice as described previously [17,18]. The open field apparatus was a square arena 50 cm × 50 cm and 20 cm in height. Mice were individually placed in the center of the apparatus and allowed to freely explore for 5 min. Both central and peripheral zone activities were tracked by an automatic monitoring system (Noldus Ethovision 3.1, Wageningen, The Netherlands).

#### 2.10.2. Grip Strength

To assess muscle strength, the mouse was allowed to place its forelimbs on the grid of a digital grip strength meter (Bio SB In Vivo Research Instruments, Santa Barbara, CA, USA) and was then pulled backward lightly until it released its grip on the grid.

#### 2.10.3. Rotarod Test

To evaluate motor learning, coordination and balance, mice were tested in an accelerating rotarod (IITC Life Science Inc., Woodland Hills, CA, USA) as described previously [19,20]. Each mouse was individually acclimatized to the rotarod apparatus. The speed was then gradually increased from 4 to 40 rpm over 5 min. The latency to fall and the actual rotation speed level within this period were automatically recorded. Mice trained for 4 trials with 30 min intervals per day for 4 consecutive days. The average retention time and end speed were used in the analysis.

#### 2.10.4. Footprint Test

The front and hind paws of the mice were inked with red and black nontoxic ink. The mice were then allowed to walk along a 50 cm long, 10 cm wide runway on a sheet of plain paper. The distances between footprints were measured to obtain “stride” and “sway” lengths.

### 2.11. Antibodies, Reagents, and Plasmids

The following primary antibodies were used in this study: anti-Thorase (Biolegend, San Diego, CA, USA, 1:1000), anti-phosphorylated S129 α-synuclein (Wako, Osaka, Japan, pSyn#64, 1:1000), anti-α-synuclein Phospho (Ser129) antibody (Covance, Princeton, NJ, USA, #MMS-5091, 1:1000), rabbit monoclonal phosphorylated S129 α-synuclein and mouse monoclonal human α-synuclein (Sigma-Aldrich, St. Louis, MO, USA, 1:1000), anti-α-synuclein (phospho S129) antibody (Abcam, Cambridge, UK, ab51253, 1:1000), mouse monoclonal α-synuclein (BD, Franklin Lakes, NJ, USA, 1:1000), anti-α-synuclein antibody (Abcam, Cambridge, UK, ab3309, 1:1000), rabbit polyclonal TH (Novus, Englewood, CO, USA, 1:1000), mouse monoclonal GFP (CST, Danvers, MA, USA, 1:1000), mouse monoclonal Myc (CST, Danvers, MA, USA, 1:1000), mouse monoclonal GAPDH (Abmart, Shanghai, China, 1:1000), rat polyclonal NeuN (Millipore, Darmstadt, Germany, 1:1000), and mouse monoclonal β-tubulin and β-actin (BOSTER, Wuhan, China, 1:2000), rabbit polyclonal SQSTM/p62 antibody (MBL, Tokyo, Japan, 1:2000), anti-HA Tag Mouse mAb (Abmart, Shanghai, China, T62939S, 1:1000).

The following secondary antibodies were used in this study: HRP anti-mouse/rabbit IgG secondary antibodies (ZSGB, Beijing, China, 1:5000), HRP anti-rabbit IgG kit and anti-mouse HRP kit (Vector Laboratories, Newark, CA, USA) and Alexa Fluor^®^ 488, 555, 647 fluorescence secondary antibodies (Thermo Fisher Scientific, Waltham, MA, USA, 1:2000).

The following reagents were used in this study: DAB kit (ZSGB, Beijing, China), Prolong Gold Antifade Mountant (Thermo Fisher Scientific, Waltham, MA, USA), QuikChange Lightning Site-Directed Mutagenesis kit (Agilent Technologies, Santa Clara, CA, USA).

Plasmids: the pRK172 plasmid harboring the gene for full-length human α-synuclein was purchased from Addgene. cFUGW and HA-ubiquitin constructs were gifted by Shiyou Li at Beijing Institute of Genome Research, Chinese Academy of Sciences. cFUGW-Thorase-GFP, cFUGW-Thorase-myc, cFUGW-α-syn-GFP, and cFUGW-α-syn (A53T)-GFP were constructed from cFUGW-GFP.

### 2.12. Statistical Analysis

Statistical analyses were performed using GraphPad Prism version 8 for Windows (GraphPad Software, La Jolla, CA, USA). Data were assessed for normality using the nonparametric *t*-test. Statistical significance was determined using one-way analysis of variance (ANOVA) and unpaired two-tailed Student’s *t*-tests.

## 3. Results

### 3.1. Thorase Conditional Knockout (cKO) Mice Exhibits Motor Dysfunction Behavior

Our previous studies demonstrate that Thorase is neuroprotective and regulates synaptic plasticity, learning, and memory [5,6,16]. Conventional Thorase knockout (KO) mice die from postnatal days 19 to 25 [5]. To avoid the lethality of Thorase ablation in mice, Thorase cKO mice were generated through the mating of CaMKIIα-iCre mice with floxed-Thorase mice, deleting Thorase in the brain except for in the cerebellum region in Thorase^floxp/floxp^ mice [5]. In addition to the impairment of short memory, Thorase cKO mice also exhibited progressive motor behavioral deficits beginning at around 5 months of age, including abnormal limb-clasping reflexes and tremors during the tail suspension test (Figure 1A). Additionally, at the age of 6 months, about 30% of Thorase mutant mice walked on their tiptoes (Figure 1B). The open field test showed that Thorase cKO mice moved slightly slower than their wild-type (WT) littermates (Figure 1C–E). The cKO mice preferred to move along the margins of the open field and spent a reduced amount of time in the center (Figure 1C,E), consistent with previous findings [5]. Then, the gait of the Thorase cKO mice displayed a reduced stride length (Figure 1F–J). The sway length was also significantly reduced in Thorase cKO mice (Figure 1F,I,J). Therefore, we proposed that Thorase might be involved in PD. Consequently, we assessed the effects of Thorase deletion on grip strength and motor coordination. Consistent with our hypothesis, during the rotarod test, Thorase cKO mice displayed significantly decreased grip strength (Figure 1K) and reduced latency to fall times and distances compared to their WT littermates (Figure 1L–N). Collectively, these findings suggest that Thorase deficiency causes phenotypes that motor behavioral deficits mimicking PD.

### 3.2. Thorase Deficiency Results in Extensive α-Synucleinopathy and Reduced TH+ Dopaminergic Neurons

Given that the behavioral deficits in Thorase cKO mice phenocopy the clinical symptoms of PD, we examined the brains of Thorase cKO mice for signs of α-synucleinopathy. Thorase cKO mice exhibited substantial α-synuclein (α-syn) accumulation that was co-labeled with high levels of phosphorylated (Ser129) α-syn (pS129-α-syn), the most frequently modified form of α-syn within PD pathological inclusions, and pathogenic fibrillary aggregates (Figure 2A). Stereological quantification analysis showed that the intensities of pS129-α-syn staining were significantly increased in the substantia nigra pars compacta (SNpc), and cortical and hippocampal regions of Thorase cKO mice compared to those of their WT littermates (Figure 2B–D), which showed a very small amount of pS129-α-syn under physiological condition and was consistent previous studies [21,22]. Stereological counting of tyrosine hydroxylase (TH) staining showed a marked reduction in TH+ fiber density in the striatum (Figure 2E,F) and a reduced number of TH+ dopaminergic neurons in the SNpc of Thorase cKO mice (Figure 2G,H). Next, we performed a Western blot assay and found that Thorase cKO mice exhibited significantly increased levels of pS129-α-syn (Figure 2I,J), which are more neurotoxic [23]. Additionally, the level of SQSTM1/p62, an abundant constituent in synuclein inclusions [24], was significantly increased (Figure 2K,L). We also found that the level of total α-syn and TX-insoluble α-syn was significantly increased in the brains of Thorase cKO mice compared to that of their WT littermates (Figure 2M–P). Collectively, these results demonstrate that Thorase deletion in the brain results in extensive α-synucleinopathy.

### 3.3. Thorase Deficiency Accelerates α-Synucleinopathy and Behavioral Impairments in a Familial PD A53T Mouse Model

To further address whether Thorase has effects on the pathogenesis and progression of PD, we examined the role of Thorase in a familial PD A53T mouse model. The hA53T α-syn G2-3 Tg mice exhibit progressive α-syn accumulation in the cerebellum, brainstem, cortex, and spinal cord, accompanied by neurodegenerative phenotypes including hyperactivity in the early stage (about 5 months), substantial α-syn pathology, and glial activation at the onset of paralysis (about 9 months) followed by death (9 to 12 months) [13,25]. Thorase cKO mice were crossbred with transgenic PD mice bearing the human α-synuclein gene with the A53T mutation (hA53T) mice. The levels of pS129-α-syn immunoreactivity in the SNpc, cortical and hippocampal regions of the Thorase cKO-hA53T α-syn (cKO-A53T) mice were significantly increased compared to those in the SNpc, cortical and hippocampal regions of the A53T littermate mice (Figure 3A–D), which is also verified by Western blot analysis (Figure 3E,F). Compared to littermate A53T mice, cKO-A53T mice also displayed a significant reduction in TH+ fiber density and a reduced number of TH+ dopaminergic neurons in striatum regions (Figure 3G,H) and in the SNpc (Figure 3I,J). The cKO-A53T mice exhibited a similar pattern of behavioral impairments in the open field test as the cKO mice (Figure 3K–M). In addition, the conditional ablation of Thorase significantly exacerbated the impairments in grip strength, motor coordination, and balance in A53T mice (Figure 3N–P).

### 3.4. Thorase Interacts with α-syn and Regulates the Degradation of Ubiquitinated α-syn

We next sought to assess whether Thorase directly interacts with α-syn and whether the accumulation of α-syn in the brains of Thorase KO mice results from a blockage in the degradation of ubiquitinated α-syn. Coimmunoprecipitation in HEK293 cells co-transfected with Thorase-myc and α-syn-GFP indicated that Thorase interacts with α-syn (Figure 4A). We also confirmed the interaction of endogenous Thorase with α-syn in mouse brain tissues (Figure 4B). As Thorase cKO mice exhibited substantial α-synucleinopathy in the brain, we hypothesized that the neurons lacking Thorase would display reduced degradation of α-syn. Therefore, HA-tagged ubiquitin (Ub-HA) and α-syn-GFP expression constructs were co-transfected with/without Thorase-myc into HEK293 cells. Immunoprecipitation assays showed that Thorase-myc overexpression significantly reduced the level of the high-molecular-weight smear representing polyubiquitinated α-syn-GFP (Figure 4C,D). Treatment with an autophagy inhibitor bafilomycin A1 (BafA1) blocked Thorase-mediated degradation of polyubiquitinated α-syn-GFP. In contrast, treatment with MG132, an inhibitor of the ubiquitin–proteasome system [26], did not prevent Thorase-mediated polyubiquitinated α-syn-GFP degradation (Figure 4C,D). These results suggest that Thorase might regulate α-syn degradation mainly through the autophagy pathway.

### 3.5. Thorase Deficiency Promotes α-syn Aggregation in Primary Cultured Neurons

Previous studies have shown that synthetic preformed α-syn fibrils (PFFs) seeded in primary neurons recruit endogenous mouse α-syn resulting in a Lewy body/Lewy neurite (LB/LN)-like pathology [15,27]. Thus, we examined the role of Thorase on the formation α-syn aggregates in primary cultured neurons. Exogenous human α-syn-A53T-PFFs were generated from recombinant α-syn-A53T and applied to primary hippocampal neurons derived from Thorase KO mice and WT littermates after in vitro culture. α-syn aggregates in WT and KO neurons were examined by staining endogenous mouse α-syn (total α-syn) with a Syn mAb. Compared to WT neurons, the Thorase KO neurons showed an increase in the extent of α-syn aggregates either at the basal level (PBS-treated) or after PFF treatment (Figure 5A,B).

To investigate α-syn recruitment to pathologic inclusions, as demonstrated by extensive phosphorylation at Ser129 (pS129-α-syn) [27], we used pSyn#64 monoclonal antibodies specific for pS129-α-syn and found that pS129-α-syn-positive neuritic and perikaryal inclusions were significantly increased in PFF-treated Thorase KO neurons compared to WT neurons (Figure 5C,D). These results further indicate an increased formation of pathologic α-syn aggregates in Thorase KO neurons.

### 3.6. Thorase Overexpression Prevents α-Synucleinopathy in PD Mouse Model A53T Mice

To validate whether Thorase overexpression attenuates α-syn accumulation in vivo, A53T transgenic mice were crossed with inducible tetO-Thorase transgenic (cTg) mice to overexpress Thorase in the forebrains [6]. Immunohistochemical staining of brains from 9-month-old mice showed that pS129-α-syn accumulation was significantly lower in Thorase cTg-A53T mice than A53T mice (Figure 6A,B). Western blot analyses also showed that hSCNA and pS129-α-syn accumulation was significantly reduced in Thorase cTg-A53T mouse brain lysates compared to that in A53T mouse brain lysates (Figure 6C,D). We also discovered that Thorase overexpression significantly prevented the accumulation of the PD-associated insoluble α-syn and pS129-α-syn fractions; soluble α-syn levels dropped only slightly but showed no significant differences (Figure 6E,F). Together, these findings reveal that Thorase overexpression attenuates α-syn accumulation in vivo.

## 4. Discussion

PD is the second most common neurodegenerative disease that currently affects 1% of people over 65 years of age [28]. Pathologically, a hallmark of the disease is the presence of intracytoplasmic inclusions called Lewy bodies that contain extensive α-synuclein in the brain [29]. The results presented here demonstrate that genetic deletion of Thorase causes extensive α-synucleinopathy in the brain and PD-like behaviors including tremors, walking on tiptoes, and impaired grip strength, motor coordination and balance. Furthermore, Thorase deficiency accelerates α-synucleinopathy and behavioral defects in the A53T mouse model of familial PD, while overexpression of Thorase prevents α-syn accumulation in A53T mice. These findings demonstrate that Thorase plays a pivotal role in α-syn metabolism and PD pathogenesis.

Thorase belongs to the AAA (ATPases associated with various cellular activities) ATPase family, containing a large group of functionally diverse enzymes that induce conformational changes in a wide range of substrate proteins. The AAA proteins function as molecular switches in the regulation of a wide variety of cellular functions, including vesicle transport, organelle assembly, membrane dynamics, protein complex assembly or disassembly, and protein unfolding and degradation [30].

Recent studies demonstrate that Msp1, an ortholog of Thorase in yeast, promoted the extraction and degradation of mislocalized tail-anchored (TA) proteins from the mitochondrial outer membrane [31], maintaining the proteostasis in mitochondria. The interaction of Cis1 with Msp1 was required for MitoCPR, a surveillance pathway that protects mitochondria in response to protein import stress [32]. Mitochondrial dysfunction is well-believed to play a crucial role in the underlying mechanisms contributing to neurodegeneration in PD [33]. Recent evidence suggests that aggregated α-syn proteins attach to the mitochondrial outer membrane and cause mitochondrial dysfunction, which contributes to oxidative stress-mediated apoptosis in neurons [34]. Thorase may serve as a membrane protein extraction machine to remove mislocalized α-syn proteins from the mitochondrial outer membrane [31], maintaining mitochondrial quality control and playing a protective role in α-synuclein-mediated neurodegeneration. However, the exact mechanism by which Thorase regulates the degradation of aggregated α-syn needs further investigation. Therefore, our future study will focus on exploring the mechanism underlying the role of Thorase in regulating mitochondrial quality control or MitoCPR.

In this study, we also verified that Thorase interacted with α-syn and regulated α-syn degradation. Treatment with autophagy inhibitor BafA1 but not ubiquitin–proteasome system inhibitor MG132 blocked Thorase-mediated degradation of ubiquitinated α-syn indicating that Thorase might promote the degradation of ubiquitinated α-syn through mitophagy other than the proteasome pathway. In addition, Thorase deficiency facilitated the formation of α-syn aggregates induced by recombinant α-syn-A53T-PFFs in Thorase KO neurons, which supports the finding of a blockage of α-syn degradation in Thorase KO neurons. Importantly, we found that Thorase overexpression prevented α-synucleinopathy in a familial PD model. Our results strongly support the important role of Thorase signaling in preventing α-synucleinopathy.

In summary, we identified Thorase as a novel key regulator of α-syn metabolism, which provides us a further understanding of the pathogenesis of α-synucleinopathies. Our findings indicate that augmentation of the level of the Thorase ATPase expression or its activity might be an attractive target for therapeutic intervention for α-synucleinopathies including PD.

## Figures and Tables

**Figure 1 cells-11-02990-f001:**
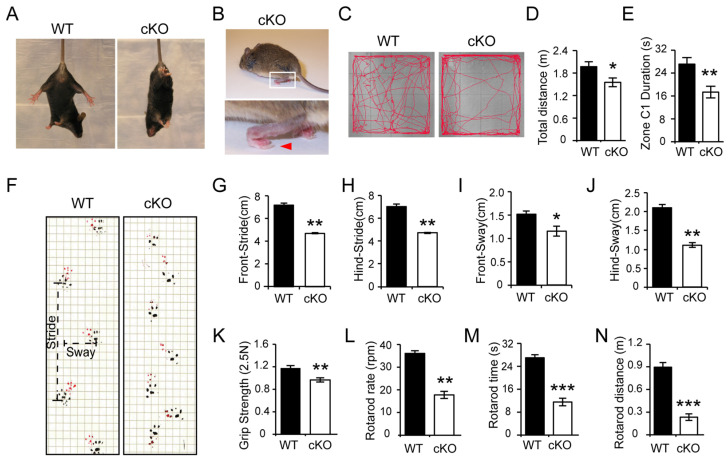
Thorase cKO mice exhibit behavioral deficits. (**A**) Thorase cKO mice show tremor and limb-clasping reflexes in the tail suspension test. (**B**) Thorase cKO mice at the age of 6 months walk on their tiptoes (red arrow). (**C**) Representative image showing tracks of Thorase cKO (n = 13) and WT littermate mice (n = 15) at the age of 5–6 months in open field tests. (**D**,**E**) Assessment of the total travel distance (**D**) and the time spent in the center (C1) (**E**) during open field tests. (**F**) Typical front (red) and hind (black) paw footprints of Thorase cKO (n = 5) and WT littermate mice (n = 4) at the age of 5–6 months. (**G**–**J**) Quantification of the stride length and sway distance as assessed by front stride length (**G**), hind stride length (**H**), front sway length (**I**), and hind sway length at the age of 5–6 months (**J**). (**K**) Assessment of the grip strength performance. (**L**–**N**) Motor coordination was assessed by the rotarod rate (**L**), the retention time on the rotarod (**M**), and the rotarod distance (**N**) for Thorase cKO (n = 13) and WT littermate mice (n = 15) at the age of 5 months. Data are presented as the mean ± SEM determined by unpaired two-tailed Student’s *t*-test. * *p* < 0.05, ** *p* < 0.01 and *** *p* < 0.001.

**Figure 2 cells-11-02990-f002:**
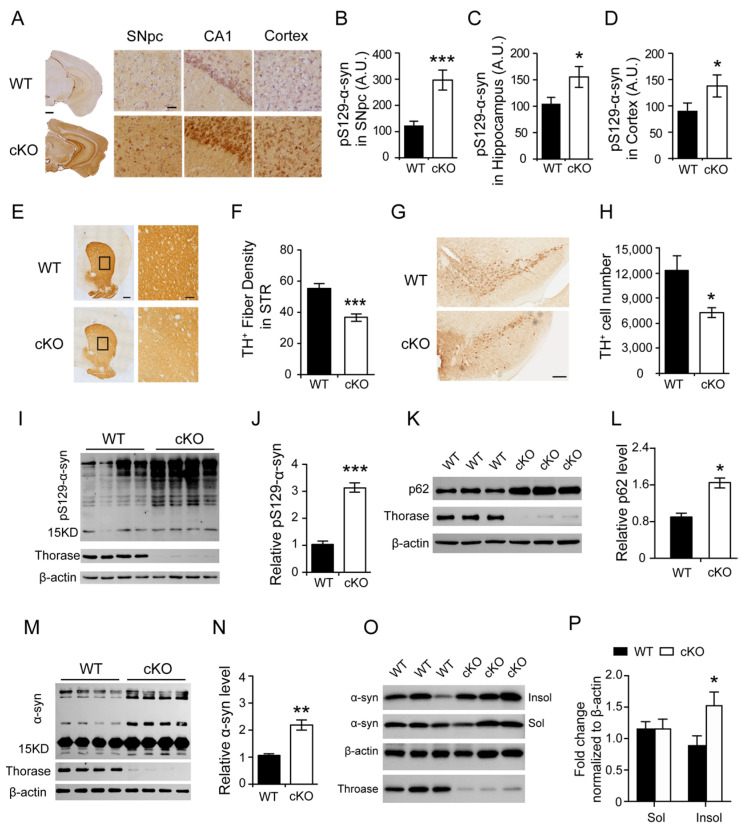
Thorase cKO mice exhibit extensive α-syn accumulation in the brain. (**A**) Immunohistochemical staining of pS129-α-syn in the brains of Thorase cKO and WT littermate mice at the age of 5 months. Scale bar, 1 mm. High-powered view of pS129-α-syn immune reactivity in the substantia nigra (SNpc), hippocampal CA1 region (CA1) and cortex. Scale bar, 25 μm. Anti-α-syn (phospho S129) antibody (Abcam, 1:1000). (**B**–**D**) Quantified pS129-α-syn optical density (OD) in the SNpc (**B**), hippocampus (**C**) and cortex (**D**) (n = 6). (**E**) Representative images of TH immunostaining using anti-TH antibody (Novus, 1:1000) in striatum region (STR) sections from Thorase cKO and WT littermate mice at the age of 5 months. Black rectangles represent for the corresponding right images of high magnification. Scale bars, 500 μm for low magnification and 100 μm for high magnification. (**F**) Quantification of TH+ fibers in the STR from Thorase cKO and WT littermate mice (n = 4). (**G**) Representative images of TH immunostaining using anti-TH antibody (Novus, 1:1000) in substantia nigra pars compacta (SNpc) regions in Thorase cKO and WT littermate mice at the age of five months. Scale bar, 200 μm. (**H**) Quantification of TH+ neurons in the SNpc regions of Thorase cKO and WT littermate mice from three independent experiments (n = 4). (**I**) Western blots showing the detection of pS129-α-syn levels in the brains of Thorase cKO and WT littermate mice. Anti-α-syn (phospho S129) antibody (Abcam, 1:1000); anti-Thorase (Biolegend, 1:1000); anti-β-actin antibody (BOSTER, 1:2000). (**J**) Quantification of the integral optical density (IOD) of pS129-α-syn normalized to β-actin (n = 10). (**K**) Western blots showing the detection of p62 levels in the brains of Thorase cKO and WT littermate mice. Anti-SQSTM/p62 antibody (MBL, 1:2000). (**L**) Quantification of the integral optical density (IOD) of p62 normalized to β-actin (n = 8). (**M**) Western blots showing α-syn in the brains of Thorase cKO and WT littermate mice at the age of 5 months. Anti-α-syn antibody (BD, 1:1000). (**N**) Quantification of α-syn normalized to β-actin experiments (n = 6). (**O**) Western blots showing the detection of TX-insoluble α-syn and TX-soluble α-syn levels in the brains of Thorase cKO and WT littermate mice. Anti-α-syn antibody (BD, 1:1000). (**P**) Quantification of the integral optical density (IOD) of Triton X-insoluble α-syn and Triton X-soluble α-syn normalized to β-actin (n = 10). Data are presented as the mean ± SEM, determined by unpaired two-tailed Student’s *t*-test. * *p* < 0.05, ** *p* < 0.01 and *** *p* < 0.001.

**Figure 3 cells-11-02990-f003:**
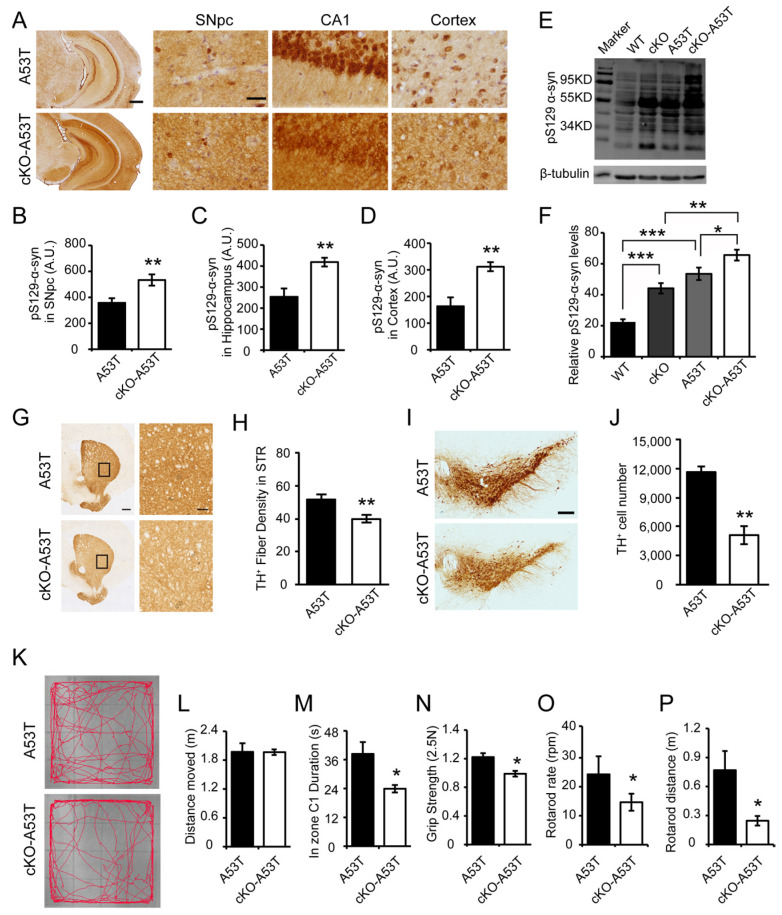
Genetic deletion of Thorase in the brain exacerbates α-synucleinopathy and behavioral impairments in PD A53T mice. (**A**) Representative images of pS129-α-syn immunohistochemical staining in the SNpc, hippocampus and cortex in sections from Thorase cKO-hA53T α-syn (cKO-A53T) and A53T littermate mice at the age of 5 months. Scale bars, 1 mm for low magnification and 15 μm for high magnification. Anti-α-syn (phospho S129) antibody (Abcam, 1:1000). (**B**–**D**) Quantification of the immunohistochemical staining of pS129 α-syn in sections of the SNpc (**B**), hippocampus (**C**) and cortex (**D**) from three independent experiments (n = 5). (**E**) Western blot assay of pS129-α-syn in brain lysates from 5-month-old Thorase cKO, Thorase cKO-A53T and age-matched littermate control mice. Anti-α-syn (phospho S129) antibody (Abcam, 1:1000), anti-β-tubulin antibody (BOSTER, 1:2000). (**F**) Quantification of pS129-α-syn levels in the Western blot assay in panel (**E**) with normalization to the level of β-tubulin (n = 5). (**G**) Representative images of TH immunostaining using anti-TH antibody (Novus, 1:1000) in STR sections from Thorase cKO-A53T and A53T littermate mice at the age of five months. Black rectangles represent for the corresponding right images of high magnification. Scale bars, 1 mm for low magnification and 50 μm for high magnification. (**H**) Quantification of TH+ fibers in the STR from Thorase cKO-A53T and A53T littermate mice (n = 4). (**I**) Representative images of TH immunostaining using anti-TH antibody (Novus, 1:1000) in SNpc regions from Thorase cKO-A53T and A53T littermate mice at the age of 5 months. Scale bar, 200 μm. (**J**) Quantification of TH+ neurons in the SNpc regions from Thorase cKO-A53T and A53T littermate mice (n = 4). (**K**) Representative images of Thorase cKO-A53T (n = 13) and A53T littermate (n = 15) mice track at the age of 5–6 months in open field tests. (**L**,**M**) Assessment of the total travel distance (**L**) and the time spent in the center (C1) (**M**) during the open field tests. (**N**) Assessment of grip strength performance. (**O**,**P**) Motor coordination was assessed by the rotarod rate (**O**) and the rotarod distance (**P**). Data are presented as the mean ± SEM, determined by unpaired two-tailed Student’s *t*-test or one-way ANOVA with Tukey’s post hoc test. * *p* < 0.05, ** *p* < 0.01 and *** *p* < 0.001.

**Figure 4 cells-11-02990-f004:**
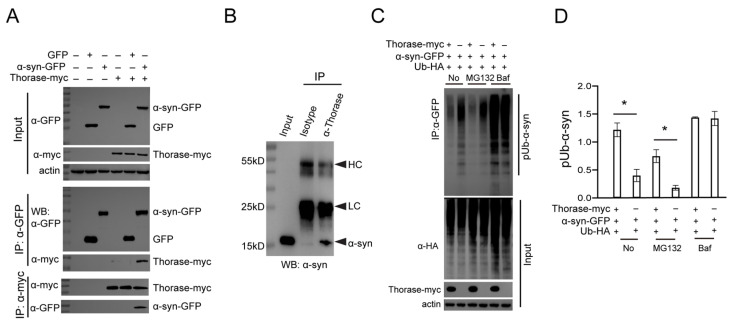
Thorase interacts with α-syn and regulates the degradation of ubiquitinated α-syn. (**A**) Coimmunoprecipitation (Co-IP) assay showing that Thorase interacts with α-syn. Thorase-myc and α-syn-GFP or GFP expression constructs were cotransfected into HEK293 cells. After 48 h, cell lysates were separated into two aliquots for immunoprecipitation assays with either an anti-GFP (CST, 1:1000) or an anti-myc antibody (CST, 1:1000). Western blotting was performed to examine the interaction of Thorase with α-syn by the use of an anti-myc or anti-GFP antibody. (**B**) Co-IP assay showing that endogenous Thorase interacts with endogenous α-syn in mouse brain lysates. Mouse isotype-IgG served as a negative control for the Co-IP experiment. Anti-α-syn antibody (BD, 1:1000). (**C**) Thorase promotes the degradation of ubiquitinated α-syn. Ub-HA and α-syn-GFP expression constructs were cotransfected with or without Thorase-myc into HEK293 cells. After 24 h, ubiquitinated α-syn was examined by Western blot. Baf was used as a negative control. Anti-HA antibody (Abmart, 1:1000); anti-Thorase (Biolegend, 1:1000); anti-β-actin antibody (BOSTER, 1:2000). (**D**) Quantification of the level of the high-molecular-weight smear representing polyubiquitinated α-syn-GFP shown in panel (**C**). Data are presented as the mean ± SEM, determined by unpaired two-tailed Student’s *t*-test. * *p* < 0.05.

**Figure 5 cells-11-02990-f005:**
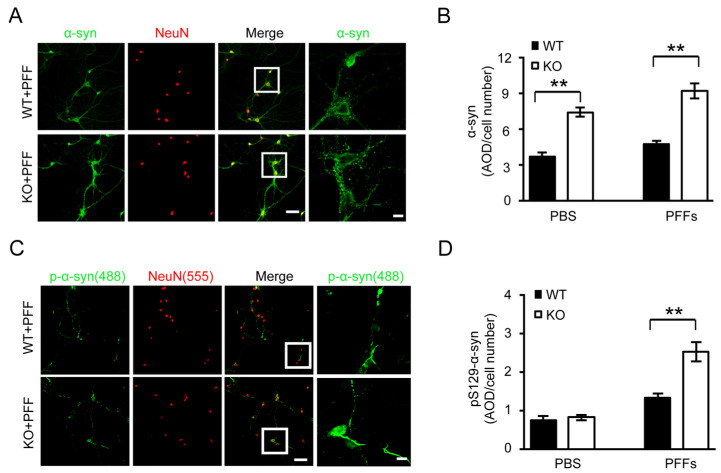
Thorase deficiency promotes α-syn aggregation in primary cultured neurons. (**A**) Immunofluorescence visualization of PFF-induced endogenous mouse α-syn aggregate formation in WT and KO neurons. Neurons were fixed in 4% paraformaldehyde and 4% sucrose before immunofluorescence staining with an anti-α-syn antibody (BD, 1:1000) for endogenous mouse α-syn (**green**) aggregates and an antibody for the neuronal marker NeuN (**red**) antibody (Millipore, 1:1000). White rectangles represent for the corresponding right images of high magnification. Scale bars, 50 μm. (**B**) Quantification of endogenous mouse total α-syn aggregates in WT and KO neurons shown in panel (**A**). (**C**) Immunofluorescence visualization of PFF-induced pS129-α-syn aggregate formation in WT and KO neurons with a pSyn#64 antibody (Wako, 1:1000) for pS129-α-syn (green) and an anti-NeuN (red) antibody (Millipore, 1:1000). White rectangles represent for the corresponding right images of high magnification. Scale bars, 50 μm. (**D**) Quantification of the immunofluorescence signal for pS129-α-syn aggregates in WT and KO neurons shown in panel (**C**). Data are presented as the mean ± SEM, determined by unpaired two-tailed Student’s *t*-test. ** *p* < 0.01.

**Figure 6 cells-11-02990-f006:**
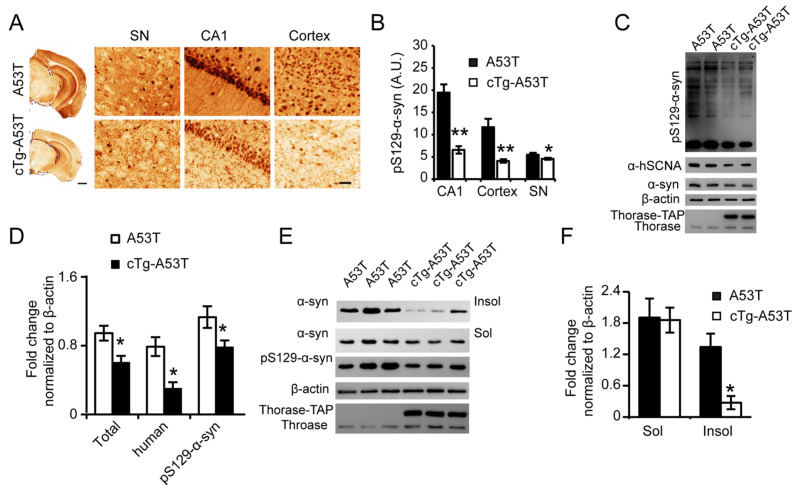
Thorase overexpression prevents α-synucleinopathy in PD mouse model A53T mice. (**A**) Representative images of immunohistochemical staining for pS129-α-syn in the SNpc, hippocampus, and cortex in sections from Thorase-TAP-hA53T α-syn (cTg-A53T) and A53T littermate mice at the age of 9 months. Scale bars, 1 mm for low magnification and 15 μm for high magnification images. Anti-α-syn (phospho S129) antibody (Abcam, 1:1000). (**B**) Quantification of the immunohistochemical staining of pS129-α-syn in the hippocampus, cortex, and SNpc. (**C**) Western blot assay of pS129-α-syn in the brains of Thorase cTg-A53T and age-matched A53T littermate mice. Anti-α-syn (phospho S129) antibody (Abcam, 1:1000); anti-α-syn antibody (BD, 1:1000); anti-α-syn antibody (Abcam, 1:1000). (**D**) Quantification of pS129-α-syn levels in panel (**C**) with normalization to β-actin (n = 5). (**E**) Western blot assay of insoluble α-syn, soluble α-syn, and pS129-α-syn in the brains of Thorase cTg-A53T (n = 6) and A53T littermate mice (n = 5). Anti-α-syn (phospho S129) antibody (Abcam, 1:1000); anti-α-syn antibody (BD, 1:1000). (**F**) Quantification of the Western blot data shown in panel (**E**). Data are presented as the mean ± SEM, determined by unpaired two-tailed Student’s *t*-test. * *p* < 0.05, ** *p* < 0.01.

## Data Availability

Not applicable.

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
