# Peer review of "ATPase Thorase Deficiency Causes α-Synucleinopathy and Parkinson’s Disease-like Behavior"

_cells, 2022, doi:10.3390/cells11192990_

Round 1
Reviewer 1 Report (Previous Reviewer 1)
The authors have addressed all my questions.
Author Response
Dear Reviewer,
Thank you very much for your efforts for the quality improvement of our manuscript. You are highly appreciated!
Best regards,
Jianmin
Reviewer 2 Report (Previous Reviewer 3)
RE: Gao et al.
The authors have made significant revisions to the text and clarified several major points. I am satisfied with their response. I have only two comments, which were also indicated in my first report, that would be of benefit to the readers:
1. Please add antibody and dilution in the corresponding Figure legend
2. Section 3.3. PD A53T Mouse Model: Please add few sentences on the distribution of alpha-synuclein pathology in different brain regions of A53T mice in relation to their age and movement deficits.
Author Response
Dear Reviewer:
We appreciate your valuable comments for improving the quality of our manuscript.
Accordingly, we have revised the manuscript with all changes highlighted in Red. Here, we provide a point-by-point response to all comments.
Comment: The authors have made significant revisions to the text and clarified several major points. I am satisfied with their response. I have only two comments, which were also indicated in my first report, that would be of benefit to the readers:
- Please add antibody and dilution in the corresponding Figure legend
Response: Thanks again for your comments. Accordingly, in the revised manuscript we have added the information about the antibodies and their dilution in the corresponding Figure legend.
Comment: 2. Section 3.3. PD A53T Mouse Model: Please add few sentences on the distribution of alpha-synuclein pathology in different brain regions of A53T mice in relation to their age and movement deficits.
Response: According to the suggestion, we have added more information about the PD A53T mouse model and cited the original paper with the details about the hA53T α-syn G2-3 Tg mice.
In addition, we have also carefully checked the English language and style.
Thanks again for your efforts for the quality improvement of our manuscript.
Best wishes,
Jianmin Zhang, Ph.D.
Professor, Department of Immunology
PUMC & CAMS
This manuscript is a resubmission of an earlier submission. The following is a list of the peer review reports and author responses from that submission.
Round 1
Reviewer 1 Report
This report describes interaction between Thorase, a AAA+ ATPase, which was previously implicated in regulation of AMPA receptor. Herein, authors show that conditional loss of Thorase in forebrain via CamK2-Cre (cKO), leads to significant motor deficits and alpha-synuclein (aS) accumulation. They also show that Those cKO also exacerbate neuropathology in an A53T mutant human aS TG mouse model (presumably M83 model). Presumably, Thorase promotes degradation of ubiquitinylated aS and loss of Thorase leads to increase in aS aggregation. In support of this, they show evidence that increased expression of Thorase, via CamK2-tTA/pTet-Thorase, prevents aS pathology in the aS Tg model. Overall, the results of manuscript is quite interesting and clearly have significance for understanding PD. However, I feel some clarification is needed.
1. Previous study on Thorase cKO mice by the Dawson lab indicate no motor deficits in Thorase cKO mice at 3-4 monthos of age. Thus, severe motor deficits seen at 4-6 months of age (Fig.1) is quite interesting. Can authors confirm that these cKO are normal at 3-4 months of age?
2. It is general view that CamKII-Cre is not active in subcortical areas, including SN. Thus, I am surprise to see loss of TH fibers and neurons (Figure 2). Can authors confirm that SN neurons show Thorase deficits in the Thorase cKO mice?
3. The increase in aS levels with loss of Thorase is quite amazing and important. However, pS129 aS may not always indicate aS pathology. I would be important to add analysis of detergent insoluble aS as an additional indicator of aS pathology.
4. The A53T TG mouse line used (M83) only show early aS pathology as a homozygous mice. It is unclear if authors are using homozygous or homozygous mice. One would not expect any pS129 aS accumulation in homozygous mice until they are vey old (>16 months). Thus, presence of significant aS pathology at 5 months of age is unexpected. Please explain. Also, predominant pathology of the M83 mouse line is in brain stem and spinal cord. Thus, analysis of these regions should be provided.
5. As Thorase cKO already show loss of SN neurons, I don't think there is increase in SN pathology as inferred by Fig 3 G-J. Similarly, behavioral impairment shown in Fig 3K-P seems comparable to what is caused by cKO alone. As with prior comment on cKO, please provide the status of Thorase expression in SN.
6. Please describe the PFF treatment of primary neurons in more detail. What was the age of the culture when PFF was added? How long was neurons exposed to PFF prior to analysis?
7. What is status of aS pathology in areas that show robust aS pathology in M83 mice?
Reviewer 2 Report
This research is about AAA+ 20 ATPase Thoras in Parkinson’s disease. The underlying mechanisms and pathogenesis of PD remains unclear, therefore this can be a one of mechanism of PD pathology. The results are very interesting. Please correct the following.
‘Behavioral Measurements’ please provide Behavioral test methods.
Please explain ‘tail suspension test’, ‘open field test’.
Figure 2A, the backgrounds of WT and cKO look different. Please match the level.
Figure 6A, please add scale bar according to ‘Scale bars, 1 mm for low magnification and 15 μm for high magnification images’ in line 375.
Reviewer 3 Report
Gao et al.
In this manuscript, the authors report that conditional knockout of Thorase (AAA+ ATPase, Atad1) in a rodent model causes defects in motor performance and abnormal posture. The authors further report that thorase deficiency is associated with accumulation of aggregated alpha-synuclein in substania nigra and other brain regions, a hallmark of synucleinopathies including Parkinson disease (PD). Further experiments in animals show that thorase deficiency exacerbates alpha-synuclein accumulation in a transgenic human alpha-synuclein A53T model, while inducible overexpression (tetO-Thoarse) lowers alpha-synuclein aggregation. Cell culture studies using transient overexpression of GFP-tagged alpha-synuclein and myc-tagged thorase indicate that the two co-immunprecipitate and that affects ubiquitination of GFP-tagged alpha-synuclein. In cultures of primary hippocampal neurons derived from thorase KO mice (unclear whether conditional or constitutive transgenic KO?) accumulate alpha-synuclein, which is further increased by exposure to pre-formed fibrillar aggregates (PFF) derived from synthetic A53T human alpha-synuclein. While the study is interesting with regards to a potential neuroprotective role of thorase in alpha-synuclenopathies and many experiments are carefully designed, I have concerns on the data presentation and lack of controls. I also recommend significant revisions to the text and authors interpretations which the authors need to address (see below).
Comments on the text
I would recommend a careful revision of the text and improving the quality. There are several statement where editing is required for example, in the abstract
Here, the AAA+ 20 ATPase Thorase was identified being essential for neuroprotective..this was shown in a previous study by some of the co-authors and not in the current manuscript. The term neuroprotective: please change to neuroprotection.
Mice lacking Thorase exhibited PD-like behavior....please modify to something like: conditional knockout of thorase resulted in motor behaviors indicative of neurodegeneration/neuronal dysfunction
Thorase interacts with α-syn and regulates the degradation 25 of ubiquitinated α-syn...please modify to biochemical and cell cultures studies presented here suggest that Thorase interacts with α-syn and regulates the degradation of ubiquitinated α-syn
Thorase deficiency promotes in the development of α-syn aggregates in primary cultured neurons...please revise promotes in the development (also in the results and discussion)
The discoveries in this study place Thorase as a novel druggable target for pharmaceutical intervention of PD.... please revise (also in the discussion) as there no data on pharmacological intervention.
Additional revisions on the some statements within the text are suggested below.
Data and Figures
Fig1. Motor behaviors in thorase cKO mice. The authors have presented results from the motor performance and ambulatory behavior and in the text mention that Thorase deficiency causes phenotypes that mimic PD-like behaviors. This statement needs to be modified. For example, hindlimb clasping defects described are not unique to rodent models of alpha-synuclenopathies and are also seen in motor neuron disease, HD and models of cerebellar ataxia.
The authors mention in the text that Thorase cKO mice also exhibited severe PD-like behavioral deficits beginning at around 5 months of age, and the data are presented with animals aged 5-6 months. However, it is confusing to see that panel B shows a mouse with tip-toe gait at 4-6 months and the text only mention that some mice exhibit this defect, the authors should specify what percentage of mice in the cohort show this behavior. Also, the text implies that defects measured in panel C-G precede defects reported in panels H-K ... Then, the gait of the Thorase cKO mice displayed a reduced stride length (Figure 1C-G)....... Consequently, we assessed the effects of Thorase deletion on grip strength and motor coordination. This also does not match the figure legend where no age is mentioned for the animal in panel D-G and the data in panel I-K are from mice between 3-5 months of age. Similarly, Fig S2 should be modified as Nissl stain does not necessary indicate loss of TH+ neurons in SNpc.
There are no data from earlier (possibly prodromal) age(s) or later progression (for example, effects on survival). This would imply that the reported deficits are spontaneous and appear suddenly. This should be clarified in the text, and if authors have performed a survival assessment in cKO mice, the data should be included. In the panels D-G, the authors should indicate how many animals were used for these tests. In case n= less than 10 animals, ideally this should be presented in the graph with each animal.
Fig2. The authors show that thorase cKO mice accumulate phosphorylated S129 alpha-synuclein in substantia nigra (SNpc), CA1 of hippocampus and cortex, which is also associated with reduced TH+ fiber density, dopaminergic neurons. I have several concerns with these data.
The IHC data in panel A, it is a bit puzzling to observe phosphorylated S129 alpha-synuclein in 5 months old WT mice, and it is also apparent that the DAB staining in cKO tissue section is increased throughout and there is a high background compared to WT section. I would recommend that the authors rule out non-specific staining (for example, skipping the primary antibody in IHC) and also probe the sections with a different phospho-S129 antibody. Hence, based on one antibody, it is hard to conclude and these controls should be included. Representative panoramic views from additional mice analyzed should also be included for the reader to appreciate that these findings are consistent in the cohorts. Please indicate antibodies used and the dilution for each antibody in the figure legend.
The data in panels B-D should be presented as actual cells counts/mm2 instead of optical density to account for the high background. The authors mention these analyses are from n=6 from 3 independent experiments. It is unclear how this was done as 3 experiments would be 2 mice per experiment to end with n=6...
Panel E-H, see note on the section co-ordinates above. It also appears that the section shown for WT is more rostral than the one shown for cKO. To account for these differences in TH+ fiber density and cell number, the authors should perform IHC on 3-4 serial sections from each animal. Cell number (panel H) should be presented as cells counts/mm2 instead of optical density. Regarding quantifications in panels F and H, the authors mention n=4 from 3 experiments. Does it mean the data represents counts 12 sections from 3 experiments? Please elaborate.
Panels I-L. Again, I am a bit puzzled to notice phosphorylated S129 alpha-synuclein in brain homogenates from young non-transgenic mice. Biochemical studies in whole brain homogenates suggest that physiologically about 4-10% of alpha-synuclein is usually phosphorylated at residue S129 in mammalian brain The authors should explain this discrepancy and indicate relevant literature (see for example Fujiwara et al. 2002 PMID: 11813001 and Anderson et al. 2006 PMID: 16847063). Again, these analyses require probing with an additional phospho-S129 antibody to validate. In the panels I and K, it is also apparent that cKO mice have increased levels of both the aggregated and total ‘monomeric’ synuclein (band around 15 kDa). If the high molecular weight bands indicate aggregated alpha-synuclein, one would expect reduction in the soluble synuclein. The authors should comment on this. Also, labeling of the SDS PAGE western blots as dimer, trimer etc for phosphorylated S129 is misleading and should be removed unless the authors show this via native PAGE or oligomer specific antibodies. Regarding quantifications in panels J and K, the authors mention n=10 or n=6 from 3 experiments. Please elaborate.
Fig3. Here the authors show that conditional thorase knockout in 5 months old A53T transgenic synucleinopathy mice increases phosphorylated S129 alpha-synuclein in SNpc, CA1 and cortex compared with transgenic A53T mice. I am little concerned since the A53T mice usually exhibit some degree of alpha-synuclein aggregation in hindbrain and spinal cord between ages 9-12 months and rarely before 6 months (unless injected with PFF alpha-synuclein; see Giasson et al, 2002 PMID 12062037 and Sacino et al 2014 PMID: 25002524). Furthermore, the brain areas shown, by the authors, to harbor significant phospho S129 staining (hippocampus, cortex and substantia nigra) are usually spared till 12 months of age. Hence, I would encourage the authors to reconsider their conclusions and attempt to rule out non-specific staining (see comment on fig2 above). In the light of data presented in figure 4 and 5, have the authors considered co-staining of thorase and phospho S129 alpha-synuclein. Minor comment: Additional markers of synuclein inclusion pathology such as p62, and gliosis would be informative.
Regarding the rest of the Figure panels in fig 3, please revise as suggested under the comments in Figure 2 on providing panoramic views with atlas co-ordinates on brain sections shown for the IHC (panels A and G) and reporting positive cell counts (panels B-D, J), removing dimer, trimer, tetramer labels in panel F and antibodies used and the dilution for each antibody in the figure legend. Please elaborate what does n=5 from 3 independent experiment mean (panel B-D).
Fig4. Minor comment: Have the author considered similar analyses using a control GFP vector or a control GFP-tagged protein coding vector as the effects on ubiquitination shown may not be specific to GFP tagged alpha-synuclein. Also, it is intriguing to notice that despite MG132 treatment, the expression levels of thorase-myc do not increase as is normally seen in conditions of transient plasmid overexpression. Does it mean that the protein has a short half-life, as this may have implications on how the protein facilitates clearance of aggregated synuclein as the authors suggest. This should be highlighted and discussed.
Fig5. Please provide details of experimental procedures of the immunofluorescence staining and settings used in the imaging. Also, the procedures provided for PFF generation and characterization are inadequate, since only a coomasie stain is provided (Fig S3). The authors should include thioflavin (or similar) dye binding and DLS analyses of the fibrils. Were the fibrils sonicated before adding to the neuronal cultures. There is no indication that the IF signal indicates alpha-synuclein aggregation as the authors have not used any specific markers such as p62 or ubiquitin in these analyses (panels A and C). Regarding quantifications (panels B and D), how were the data normalized as there is no indication of total cell counts or area. Intriguingly, the authors show that neuronal cultures from thorase cKO mice have baseline synuclein staining (PBS group, panel B) but matching levels of phospho S129 staining. This may be contrasting to what the authors show in figure 2 in the brain of thorase cKO mice. Also, additional biochemical analyses (eg western blots) should be performed to support that PFF addition in these cultures increased alpha-synuclein aggregation as the authors have done in other analyses. It is also unclear as to these data represent a single or multiple experiments. The statement in the text that LB-like and LN-like fibrillary aggregates in WT and KO neurons were examined is misleading and not supported by the data. Please rephrase.
Materials and Methods
Please see comments on Fig. 5 above reading IF analyses and PFF characterization.
Also, please move the reagents/antibodies section from the supplementary to main text. Please indicate antibody source and the dilution used.
